# An Integrative Literature Review of Interventions to Protect People with Disabilities from Domestic and Family Violence

**DOI:** 10.3390/ijerph20032145

**Published:** 2023-01-25

**Authors:** Pamela Saleme, Tori Seydel, Bo Pang, Sameer Deshpande, Joy Parkinson

**Affiliations:** 1Social Marketing @ Griffith, Department of Marketing, Griffith University, 170 Kessels Road, Nathan, QLD 4111, Australia; 2Australian eHealth Research Centre, CSIRO, Level 7 296 Herston Road, Herston, QLD 4029, Australia

**Keywords:** disability, systematic review, domestic violence, family violence, intervention, prevention

## Abstract

Purpose: While domestic and family violence against people with disabilities is an ongoing and crucial public health concern, and awareness of the extent of violence against people with disabilities is growing, research on the field is still limited. Thus, the present review aims to systematically identify and synthesize evidence and effectiveness from intervention strategies to increase the awareness and skills of those with disabilities to reduce and prevent domestic and family violence against them. Method: PRISMA guidelines were followed to perform a systematic search of seven scientific databases to identify the peer-reviewed literature. Results: A total of 17 eligible studies were identified (14 evaluations and 3 descriptive studies), with most taking place in developed countries. Children and women are the most frequent victims, and they were therefore the most common target audience of the included studies. Sexual, physical, and verbal abuse were the most reported types of abuse, while financial abuse and neglect were studied less often. Interventions also focused on a diversity of disabilities, including learning, intellectual, mental, and physical impairments. Overall, the intervention strategies reflected a substantial homogeneity: focus on training and education as well as setting up channels and facilities for victims to seek help. Nine studies yielded significant positive outcomes using various strategies and techniques, while five studies had mixed results, and three studies only reported on the intervention strategies but did not evaluate the results. Conclusions: This review confirms a significant gap in the literature on domestic and family violence against people with disabilities and how to prevent and address the violence through evidence-based interventions. Several recommendations to improve future research and practice are proposed.

## 1. Introduction

The prevalence of domestic and family violence globally reveals a crisis on a pandemic scale [1]. In emergency contexts, such as the COVID-19 pandemic, domestic and family violence can increase significantly [2]. More specifically, the pandemic magnifies the existing issues and barriers that people with disabilities who are experiencing domestic violence are facing [3]. In Australia, 5.9% of women and 5.6% of men living with a disability or a long-term health condition experienced violence in 2016 [4]. More concerning, women with a disability were twice as likely to experience domestic violence by a cohabiting partner than women without a disability [4].

The Australian Public Service Commission [5] defines ‘disability’ as sensory, intellectual, physical, or psychosocial impairments, as well as head injuries and other conditions that restrict everyday activities. Additionally, it is important to emphasise article 16 of the United Nations [6] Convention on the Rights of Persons with Disabilities that mandates that “States Parties shall take all appropriate legislative, administrative, social, educational and other measures to protect persons with disabilities, both within and outside the home, from all forms of exploitation, violence, and abuse, including their gender-based aspects.” People with disabilities face high rates of violence victimization, as documented by several studies [7,8,9]. These acts can include physical violence, coercion, control, and emotional abuse [10,11,12]. Moreover, adults and children with disabilities are significantly more likely or equally as likely, compared with those without disabilities, to experience domestic and family violence [13,14,15,16]. Family violence refers to violence between family members as well as between current or former intimate partners, and it can include acts of violence between a parent and a child, between siblings, or toward elders [17]. Domestic violence is considered a subset of family violence, where violent behaviour exists between current or former intimate partners, where one partner tries to exert power and control over the other [17]. Domestic and family violence can be exhibited in many forms, including physical violence, sexual abuse, emotional abuse, intimidation, economic deprivation, or threats of violence (Australian Institute of Health and Welfare, 2018). Domestic and family violence can occur within families by parents, partners, siblings, neighbours, and personal care givers [18]. However, extensive systematic reviews provide evidence that people with disabilities experience significantly more domestic and family violence than those without disabilities [8,19,20,21]. Moreover, the issues and risks facing people with disabilities include reliance on the perpetrator for care and assistance, barriers to reporting abuse and seeking help, fear of retaliation and other negative consequences if abuse is reported, emotional abuse related to the disability, and the exacerbation of secondary physical and mental health sequalae of abuse [3]. Additionally, several factors increase the risk of violence for people with disabilities, such as non-assertive behaviour, compliance behaviour for the caregiver’s ease, reliance on a caregiver, and lack of knowledge [22,23,24]. This observation necessitates the purposeful inclusion of all people with disabilities in evidence-based violence prevention programs and interventions [25]. 

Much of the early research on domestic and family violence experienced by people with disabilities focused on physical, psychological, and sexual abuse, particularly on women and children with intellectual disabilities [11]. Examples of other types of disability-related domestic and family violence include medication manipulation, denial of access to communication devices, refusal to assist with essential activities, and destruction of adaptive equipment; however, this is less explored in the literature [10,11,12].

While awareness of the extent of domestic and family violence against people with disabilities is growing, research on the field is still limited [26,27]. Several publications highlight the importance of better preventing violence against people with diverse disabilities across sexes and in various settings [16,25,28]. Most publications, however, focus on broad policy and legal recommendations or provide good practice suggestions. Developing programs focusing on the prevention of domestic and family violence are important as well as supporting those experiencing domestic and family violence and helping their supporters to recognize signs of the different types of domestic and family violence. In this context, the present review aims to systematically identify and synthesize evidence from strategies to reduce and prevent domestic and family violence against people with all types of disabilities, as well as strategies to support those with a disability, regarding how to recognize the types of abuse and the signs of abuse.

This study makes three important contributions: first, research addressing domestic and family violence against people with disabilities is synthesized. Second, intervention strategies used to prevent and address this issue are highlighted. Third, this study contributes insights to inform a broader call to action for addressing domestic violence against people with disabilities. Finally, several recommendations to improve future research and practice are proposed. This review aims to include people with diverse types of disabilities who may experience disabilities and interventions quite differently.

## 2. Materials and Methods 

This review aims to systematically identify domestic and family violence prevention interventions for people with disabilities and review the strategies employed and their effectiveness. The Preferred Reporting Items for Systematic Reviews and Meta-Analyses (PRISMA) guidelines [29] were followed to ensure completeness and transparency of the review process. Following the examination of previous systematic literature reviews’ search terms [20,21,26], 117 terms were identified as potentially relevant. After six tests with iterations in combinations, the following search terms were used for this review: 

Disabilit* OR disabl* OR impair* OR handicap* OR retard* (The search term *retard** was used as per previous systematic reviews [19,20])

AND

“Domestic violence” OR “domestic abuse” OR “family violence” OR “family abuse” OR “child abuse”

AND

intervention* OR evaluation OR trial OR campaign* OR program* OR experiment* OR impact* OR strateg* OR solut* OR pilot OR communicat* OR information OR awareness

Note: The asterisk allows for the inclusion of term variations (e.g., singular vs. plural).

Only publications after the year 2000 were included in the search to ensure the studies were not outdated. No limits were applied to geography to capture the broadest subset of evidence to understand the delivered approaches. 

Seven electronic scientific databases commonly used in reviews of empirical research were searched: EBSCO (all databases), Ovid (all databases), ProQuest (all databases), Taylor & Francis, Emerald, Web of Science, and Embase. Title, abstract, and keyword searches, where available, were used to maximize the coverage of potentially eligible studies. Each suitable study was reviewed independently by at least two reviewers to ensure thorough data extraction. 

All downloaded articles were imported into Endnote X9. After duplicates were removed, titles and abstracts were reviewed for eligibility. Papers were included if they met the following criteria: Research related to what is known about the risks and impact of domestic and family violence on people with disabilities;Domestic violence communication and engagement strategies and interventions;Written in English;Full text available;Peer-reviewed.

Any papers that were any of the following were excluded: Not written in English;Not peer-reviewed (e.g., newspapers, theses, or conference proceedings);Not focused on people with disabilities;Not domestic/family violence focused;Focused on disability as a consequence of abuse;Not describing an intervention, case study, or program.

Two researchers applied the inclusion and exclusion criteria across the extracted records independently. Any discrepancy between researchers was resolved through discussion and consensus with a third reviewer. Backward and forward searches from the identified studies were conducted along the process to ensure the comprehensiveness of the review.

### Data Extraction and Analysis

A total of 17 eligible studies were identified through the scientific literature search. Figure 1 summarizes the process of this review. 

Two researchers conducted data extraction from the 17 eligible intervention studies in terms of their study demographics (e.g., study location, study design, study aim), intervention details and strategies (e.g., type of abuse, type of victims, services implemented), and intervention effectiveness. 

## 3. Results

### 3.1. Study Characteristics

Overall, most studies were conducted in developed countries, i.e., the USA (n = 9), the UK (n = 3), and Australia (n = 2), with one study in Canada, one in Turkey, and one in South Korea. The study designs were cross-sectional (n = 4), cohort (n = 5), repeated measures (n = 4), quasi-experimental (n = 2), and randomized control trials (n = 2). The sample ages were diverse, depending on the target population (children and young people, women, or adults in general), and ranged from 10 to 87 years old. See Table 1 for more details.

### 3.2. Intervention Characteristics

A wide range of programs targeted various types of abuse and victims; however, with women being the most common victims, they are the most commonly targeted audience of the included studies. Studies included a range of disabilities, including learning disabilities (n = 4); intellectual disabilities (including mental retardation) (n = 6); cognitive, physical, and mental disabilities (n = 4); and all/any disability (n = 3). The types of abuse were mainly focused on sexual, physical, and verbal abuse. Financial abuse and neglect were less studied, based on the evidence gathered from this review. See Table 2 for more details.

The included studies showed homogeneity of strategic approaches. Most interventions implemented education and training in different formats to prevent domestic violence, such as group activities/support groups and class training, primarily for people with disabilities. Computer-based services for reporting and intervening in domestic and family violence were also found. Multiple studies applied multiple modes with a mixture of approaches.

Many interventions focused on women and children with intellectual disabilities, presenting a gap in the literature for programs focused on other disabilities, such as physical and psychosocial disabilities. One intervention included programs focused on men and employed a computer-based module that included realistic videos of men with disabilities sharing their experiences. Table 3 illustrates the description of the intervention strategies implemented for each type of target audience.

### 3.3. Effectiveness of Interventions

The interventions found in the sample applied various strategies and techniques to address domestic violence against people with disabilities in diverse contexts. As shown in Table 4, 53% of the studies in the sample (N = 9) had positive results after the interventions. Twenty-nine percent of the studies had mixed results (N = 5), and 18% of the studies reported on the intervention strategies but did not evaluate the results (N = 3). See Table 4 for more details.

The studies yielded positive significant outcomes using training techniques such as role-play and in-situ scenarios, behavioural skills training, individual and group face-to-face lessons, storytelling (including with a storybook), computer-based solutions, applying realistic videos to training materials, psychoeducation, carer-focused training, and family programs. The outcomes included safety skills, self-efficacy, and safety behaviours [33]; self-protective and decision-making skills [37]; abuse awareness; safety knowledge and skills; safety self-efficacy; social support; and safety promoting behaviours [32,40].

Face-to-face group lessons were frequently reported, resulting in positive outcomes in four studies [33,37,40,43]. In a small sample, Hughes et al. [33] found significant increases from baseline to postintervention on measures of self-efficacy and safety skills; furthermore, although not statistically significant, improvements were also found in safety-promoting behaviour. In a larger sample (n = 58), Hickson et al. [37] evaluated the ESCAPE-DD curriculum for increasing the effective decision-making skills of women and men with intellectual disabilities in hypothetical situations of abuse. The study found that the program was associated with an increased application of effective decision-making skills in response to scenarios involving sexual, physical, and verbal abuse. The program was also associated with increased overall and safe-now effective decision-making scores. Likewise, findings from Robinson-Whelen et al. [40] suggested that the ASAP for Women program had the potential to enhance protective factors (abuse awareness, abuse and safety knowledge, safety skills, safety self-efficacy, and social support) and safety-promoting behaviours in women with disabilities. The study found that the intervention group scored significantly better than the control group on every protective factor measured, at the post-test, six-month follow-up, or both time points. Finally, Cavalier [43] found positive effects of the New Freedom Program, an existing domestic violence group program adapted for women with disabilities. The study found that women regularly attended the sessions and reported improved understanding and confidence in addressing abusive behaviour, as well as using that knowledge in developing new relationships.

In contrast, the second most-used strategy, using realistic video scenarios or testimonials for training, reported mixed results in three studies [32,35,36] and positive effects in one study [39]. Lund et al. [39] found positive outcomes for the Men Safer and Stronger Program, which used a computer-based training solution that included a realistic video of survivor testimonies. In this pilot study, the participants regarded the program as positive, helpful, and easy to use, given their disability [39]. The participants preferred to use an internet program such as this one to learn about abuse reporting, rather than telling a professional, and for answering personal questions about abuse, rather than telling friends or family [39]. In contrast, Robinson-Whelen et al. [32] performed a randomized controlled trial evaluating the effects of a computerized disability-specific abuse assessment tool that offered an accessible and anonymous method to self-screen for sexual violence and abuse for women with disabilities. The study found that the program had a significant effect on abuse awareness but no measurable effect on safety self-efficacy or safety-promoting behaviours [32]. However, the differences in the sample between the women who had abuse experiences in the past year and those who reported little or no past year abuse (abuse status) were considerable [32]. Abuse awareness results were primarily noted among women who had little or no abuse experience (in the past year). Thus, the findings from this study support the practice of offering abuse and safety awareness information to women who are assumed to be at low risk for abuse [32].

Other effective techniques used in the sample for domestic violence training for people with disabilities included role-play, in-situ scenarios, and behavioural skills training [29,31]. Egemo-Helm et al. [31] evaluated the influence of training on adopting desired safety behaviours, including refusing to engage in risk behaviour, leaving the situation, and reporting the incident. The study found that 75% of the participants maintained their behaviours after one month and 50% after three months. Similarly, Kim [29] found that participants maintained their safety behaviours for up to 10 weeks (N = 3). 

## 4. Conclusions

This integrative review identified 17 interventions implementing strategies to improve skills and raise awareness to mitigate domestic and family violence against people with disabilities. Most studies took place in developed countries, primarily in the USA. Children and women are the most frequent victims of domestic and family violence; they are, therefore, the most common target audience of the included studies. Sexual, physical, and verbal abuse are the most reported abuse types, with less reporting of financial abuse and neglect. The interventions focused on a diversity of disabilities, including learning, intellectual, mental, and physical impairments. This review confirms a significant gap in the domestic and family violence literature regarding people with disabilities and strategies to prevent and address domestic and family violence, particularly for women. While awareness of specific problems faced by people with disabilities has been raised previously [3,8,13,16], and domestic and family violence has increasingly become an area of concern for policymakers, minimal empirical research has addressed this issue, and the literature offers limited strategy guidance [26,27]. 

The first implication of this study is that the review has synthesised the existing literature on strategies to address domestic and family violence against people with disabilities, particularly women. The intervention strategies reported reflect a strong homogeneity. Most strategies focus on training and education or setting up channels and facilities for victims to seek help. More specifically, the implementation of training and education has been customized to suit different demographics. Programs tend to implement multiple strategies to achieve many purposes, with few programs having a singular and focused approach. While implementing multiple strategies may assist in targeting the complexities of domestic and family violence for people with disabilities, this may make it difficult to attribute effectiveness to a particular strategy. Many interventions with collaborations or partnerships with various stakeholders, including local community groups, domestic violence services, and disability services, highlight the importance of these collaborations to achieve collective impact and to reach the people with disabilities needing these services and interventions. The review highlights the need for including monitoring and evaluation in program design and implementation along with larger sample sizes to enable the evaluation of strategy effectiveness. This may indicate the need for greater investment in programs targeting domestic and family violence for people with disabilities. 

The second implication of this study is the identification of a range of strategies that have been used to target domestic and family violence in people with disabilities. These include communication strategies, education, and training strategies. Communication strategies have been useful for raising awareness and changing attitudes around domestic and family violence perceptions. Organizations often use communication campaigns to convey that domestic violence is unacceptable, and these efforts range from small community-based programs to nationwide advertising campaigns [45]. Communication campaigns focus on raising awareness about what constitutes domestic and family violence in all its forms, and these campaigns highlight the available help services; this is important for addressing the outcomes of domestic and family violence in addition to assisting in its prevention. Furthermore, communication campaigns can provide advice for victims, perpetrators, professionals, and bystanders on how best to combat domestic violence, in addition to changing social attitudes, perceptions, and beliefs that normalize and trivialize domestic and family violence [45]. The third implication of this study is the need for the design of education and training programs for people with disabilities to include easy accessibility and a supportive tone. Computer-assisted abuse screening programs, similar to those identified by Lund et al. [39] that help victims identify violence, and educational programs may be effective methods of support for those with disabilities when mandatory reporting laws (for child safety) may discourage disclosure and when the goal is to increase abuse awareness for the person experiencing domestic and family violence. Furthermore, future programs may be developed using participatory or emancipatory research (knowledge that can benefit disadvantaged people) [46]. This entails including those individuals (people with a disability, significant people in children’s lives, and other stakeholders) in every aspect of an intervention design.

Domestic and family violence is a crucial public health concern that significantly affects women and people with disabilities. However, this review suggests that empirical research of intervention strategies to reduce and prevent domestic and family violence, specifically research targeting the concerns and particular context of people with disabilities, offers little guidance to practitioners and policymakers. However, insights from this review inform a broader call to action for addressing domestic and family violence against people with disabilities. Future research and program development should consider the target audience’s cognitive, intellectual, and physical abilities and focus on the interventions’ long-term impacts (e.g., safety skills). Prevention strategies targeting women and people with disabilities and their carers that increase awareness, confidence, and emotional literacy should be developed and implemented to negate some of the negative impacts of domestic and family violence. Additionally, a range of strategies should be developed that are accessible and appropriate: for example, role modelling exercises, videos, repetition, drawings, photos, and recordings to support learning and understanding. Finally, programs should focus on the safety needs of individuals with a range of disabilities beyond intellectual or developmental disabilities and include a broad range of domestic and family violence types beyond physical and sexual abuse, including, for example, coercive control and financial abuse.

This review has several limitations. First, the outcome measures of the included interventions were not considered in this study, due to the lack of evidence reported in the reviewed papers. Only the overall effectiveness of strategies employed was specified, limiting the applicability of the strategy and evaluation to future programs. Moreover, this review did not perform risk of bias assessment, reducing the data’s reliability. Third, the review included three studies that only report on the intervention’s description but do not perform empirical testing. Finally, other types of violence against people with disabilities were not included in this review, such as bullying and violence in institutions. However, future reviews should investigate these issues. 

## Figures and Tables

**Figure 1 ijerph-20-02145-f001:**
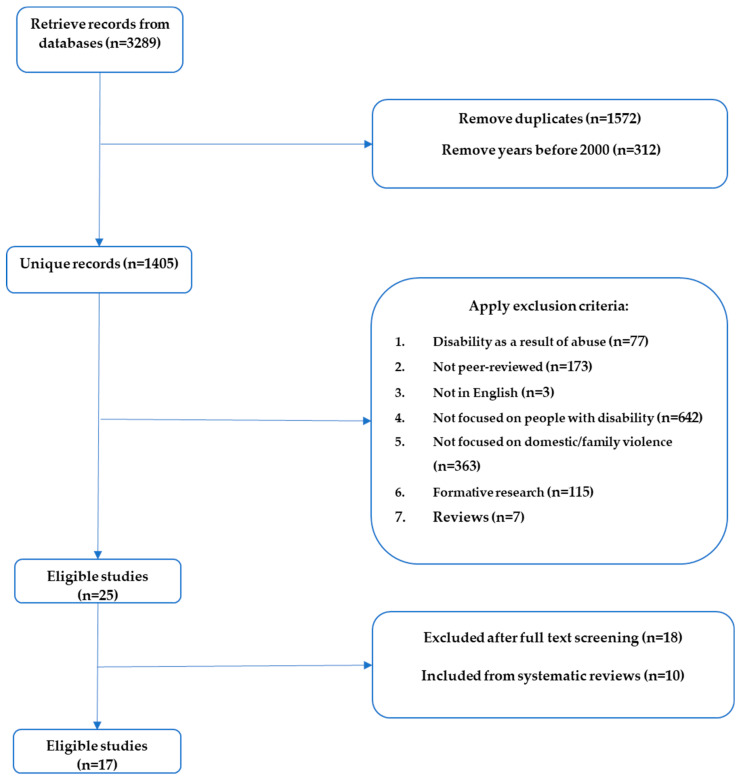
Systematic search flowchart.

**Table 1 ijerph-20-02145-t001:** Study Characteristics.

No.	Lead Author	Location	Study Design	Project Name	Sample Age
1	Kim (2016) [29]	South Korea	Cohort	N/A	11–13 years
2	Kucuk et al. (2017) [30]	Turkey	Cohort	N/A	10–14 years
3	Egemo-Helm et al. (2007) [31]	USA	Cohort	N/A	34–47 years
4	(Robinson-Whelen et al., 2010) [32]	Texas, Washington, Oregon, USA	RCT	Safer & Stronger Program (SSP)	Middle-aged women
5	Hughes et al. (2010) [33]	USA	Cohort	A Safety Awareness Program for Women (ASAP for Women)	40–62 years
6	Barber et al. (2000) [34]	UK	Repeated measures	N/A	20–33 years
7	Mazzucchelli (2001) [35]	Perth, Australia	Quasi-experimental	Feel Safe	M = 31 years
8	Peckham et al. (2007) [36]	Northumberland region (Ontario, Canada?)	Repeated measures	N/A	N/A
9	Hickson et al. (2015) [37]	USA	Repeated measures	ESCAPE-DD	M = 38.81 years
10	Lund and Hammond (2014) [38]	USA	Cross-sectional	Stopping Abuse For Everyone (SAFE)	N/A
11	Lund et al. (2015) [39]	USA	Cross-sectional	Men’s Safer and Stronger Program (SSP for Men)	20–64 years
12	Robinson-Whelen et al. (2014) [40]	USA	RCT	A Safety Awareness Program for Women (ASAP for Women)	18–87 years (M = 47.79)
13	Dryden et al. (2017) [22]	Boston, USA	Repeated measures	IMPACT	17 years
14	Cramer et al. (2013) [41]	Virginia, USA	Quasi-experimental	Interactive Community Assistance Network (I-CAN!)	N/A
15	Collins and Walford (2008) [42]	Wales, UK	Cross-sectional	Keeping Safe	N/A
16	Cavalier (2019) [43]	Bristol, UK	Cohort	Freedom Program	N/A
17	Baldry et al. (2005) [44]	NSW, Australia	Cross-sectional	N/A	N/A

**Table 2 ijerph-20-02145-t002:** Intervention details.

No.	Lead Author	Type of Abuse	Type of Victim	Type of Disability
1	Kim (2016) [29]	Sexual abuse	Children	Mild to moderate learning disability
2	Kucuk et al. (2017) [30]	Sexual abuse	Children	Mild mental retardation
3	Egemo-Helm et al. (2007) [31]	Sexual abuse	Women	Mild to moderate mental retardation
4	(Robinson-Whelen et al., 2010) [32]	Interpersonal violence	Women	Any disability consistent with the Americans with Disabilities Act (1990) (includes deafness)
5	Hughes et al. (2010) [33]	Interpersonal violence	Women	Cognitive, physical, and speech disabilities
6	Barber et al. (2000) [34]	Sexual abuse and assault	Women	Mild to moderate learning disability
7	Mazzucchelli (2001) [35]	Domestic violence	Men and women	Intellectual disability
8	Peckham et al. (2007) [36]	Sexual abuse	Women	Intellectual disability
9	Hickson et al. (2015) [37]	Sexual, physical, and psychological abuse	Men and women	Intellectual disability
10	Lund and Hammond (2014) [38]	Financial, sexual, physical and verbal abuse, neglect, and victim-blaming	Men and women	Intellectual disability
11	Lund et al. (2015) [39]	Domestic violence	Men	Physical, cognitive, and mental health disability
12	Robinson-Whelen et al. (2014) [40]	Domestic violence	Women	Physical, cognitive, and mental health disability
13	Dryden et al. (2017) [22]	Maltreatment, sexual and domestic abuse	Adolescents	Cognitive or physical disability
14	Cramer et al. (2013) [41]	Domestic and sexual violence	Men and women	All types of disability
15	Collins and Walford (2008) [42]	Community safety issues and domestic violence	Men and women	Learning and mental health disability
16	Cavalier (2019) [43]	Domestic violence	Women	Learning disability
17	Baldry et al. (2005) [44]	Child abuse/neglect as high-risk indicators for families in distress	Children	All types of disability

**Table 3 ijerph-20-02145-t003:** Description of strategies per target audience.

Target Audience	Lead Author	Intervention Strategies
Children & adolescents	Kim (2016) [29]	Training and role-play assessment sessions.
	Kucuk et al. (2017) [30]	A lesson was given face-to-face with each child at the same time of the week over a total period of four weeks.
	Dryden et al. (2017) [22]	Education program for high school students. The intervention took place on-site at each school and consisted of 10–90-min weekly class sessions.
	Baldry et al. (2005) [44]	Crisis interventions. Three family-focused programs; at-home, time-flexible services.
Women	Egemo-Helm et al. (2007) [31]	Behavioural skills training (BST) program.
	Robinson-Whelen et al. (2010) [32]	A computer-based assessment tool offered an accessible and anonymous method for women with disabilities to self-screen for domestic violence.
	Robinson-Whelen et al. (2014) [40]	Weekly classes contained didactic and interactive components, including weekly action planning with group feedback and problem-solving, affirming messages, and relaxation training.
	Hughes et al. (2010) [33]	Eight 2.5-h weekly class sessions.
	Barber et al. (2000) [34]	Women’s group met in 10 weekly sessions of 2 h duration to run interactive and structured educational sessions, supplemented with supportive and non-confronting group discussion.
	Peckham et al. (2007) [36]	Educational support group for survivors and carers ran concurrently.
	Cavalier (2019) [43]	Group program regularly run throughout the country by various organizations and individuals.
Adults in general	Cramer et al. (2013) [41]	Online computer module based on empowerment and capacity building for seeking protective orders.
	Collins and Walford (2008) [42]	College staff-developed course that ran once a week over an academic year.
	Mazzucchelli (2001) [35]	Individual and group weekly learning sessions.
	Hickson et al. (2015) [37]	Small-group instructional sessions of an abuse prevention curriculum.
	Lund and Hammond (2014) [38]	Trained facilitators delivered a session on the abuse psychoeducation program.
Men	Lund et al. (2015) [39]	The user-guided program contained eight modules, including definitions and examples of abuse, risk factors for abuse, survivor narratives, and strategy suggestions designed to increase safety.

**Table 4 ijerph-20-02145-t004:** Intervention strategies and effectiveness.

	N	1	2	3	4	5	6	7	8	9	10	11	12	13	14
No.	Strategies/Techniques	Role-Play Scenarios	In-Situ Scenarios/Training	Behavioural Skills Training	Individual Face-to-Face Lesson	Group Face-to-Face Lesson	Storytelling/Storybook	Computer-Based/Online	Domestic Violence Assessment Tool	Realistic Video Scenarios/Training	Survivor/Support Group	Carer/Professional Training	Psychoeducation	Family Program	Support Services Accessibility
1	Kim (2016) [29]	+	+												
2	Kucuk et al. (2017) [30]				+		+								
3	Egemo-Helm et al. (2007) [31]	+	+	+											
4	Robinson-Whelen et al. (2010) [32]							m	m	m					
5	Hughes et al. (2010) [33]					+									
6	Barber et al. (2000) [34]										m				
7	Mazzucchelli (2001) [35]				m	m				m					
8	Peckham et al. (2007) [36]									m	m	m			
9	Hickson et al. (2015) [37]					+									
10	Lund and Hammond (2014) [38]											nr	nr		
11	Lund et al. (2015) [39]							+		+					
12	Robinson-Whelen et al. (2014) [40]					+									
13	Dryden et al. (2017) [22]	m		m											
14	Cramer et al. (2013) [41]							nr				nr			nr
15	Collins and Walford (2008) [42]			nr		nr									
16	Cavalier (2019) [43]					+									
17	Baldry et al. (2005) [44]													+	

m = mixed effects; nr = no empirical results reported; + = significant positive effects (at *p* < 0.05).

## Data Availability

Not applicable.

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
