# Peer review of "An Integrative Literature Review of Interventions to Protect People with Disabilities from Domestic and Family Violence"

_ijerph, 2023, doi:10.3390/ijerph20032145_

Round 1
Reviewer 1 Report
Thank you for the opportunity to review this research and congratulations to the authors for identifying an important gap in the literature on domestic violence. Find below a few minor changes.
Methods
The flowchart needs to be adjusted to make the contents of the boxes readable.
Results
The strategies/interventions in row 2 of the table are not visible.

Author Response
- General comments
There’s need for the authors to check and correct typographical and grammatical errors. An English editing service may be used for this. Response: We have proof read the paper. - Introduction
The existing literature is well covered and the gap to be filled with respect to people with disability well highlighted. Response: Thank you - Methods
The methods are well presented. However, the flowchart needs to be adjusted to make the contents of the boxes readable. Response: We have amended the flow chart as recommended. - Results
The results are clearly presented, but the strategies/interventions in row 2 of table 4 are not visible. Response: We have edited the tables to ensure all rows are visible. - There is a need for clearer presentation of the identified interventions and their pros and cons possibly in a table. Response: This is included in Table 4.
- Conclusion
The conclusion is too long and there is much repetition of the information presented in the introduction and results. This section needs to be shortened and focussed more on the gaps identified and suggested recommendations for filling those gaps in future research and
interventions to address domestic and family violence among people with disability. Response: Thank you, we have now shortened the conclusion section to focus on the implications of the study, call for future research and limitations.
Reviewer 2 Report
This submission is a literature review of prevention effects targeting IPV and family violence with respect to people with disabilities. This could certainly be of interest to readers of the International Journal of Environmental Research and Public Health.
Formatting: Good.
Title: Perhaps include the word “research” or “evaluations.” I had assumed that this was the case but found out later that three references were not evaluations. Most similar systematic reviews are limited to evaluated programs, and I suggest revising the current article to adhere to this standard
Abstract: Good.
Writing: Generally well-written. Some long paragraphs could be broken up for easier reading.
Introduction: Generally well-presented and well supported by the literature.
Methods: The PRISMA guidelines are well-explained.
Results: Finding 17 studies represents a strong body of work. However, why include the three studies that were descriptive only? Perhaps mention these in the introduction or elsewhere; but it dilutes the quality of the systematic review.
Discussion: Generally good.
Conclusions and Limitations: Appropriate.
APA7 References: Good but a few random errors.
· Add digital object identifiers (dois) for journal articles. This is inconsistent.
In summary, the systematic literature review was well-conceptualized and interesting. With the exception of including the non-evaluated programs, the authors compared and contrasted the available evaluated programs in the appropriate manner. This submission is valuable.
Author Response
1. Formatting: Good.
2. Title: Perhaps include the word “research” or “evaluations.” I had assumed that this was the case but found out later that three references were not evaluations. Most similar systematic reviews are limited to evaluated programs, and I suggest revising the current article to adhere to this standard. Response: We have added "integrative review" to the title and defined the included studies to match this description. New title: An Integrative Literature Review of Interventions to Protect People with Disabilities from Domestic and Family Violence
3. Abstract: Good.
4. Writing: Generally well-written. Some long paragraphs could be broken up for easier reading. Response: We have broken up long paragraphs to improve readability.
5. Introduction: Generally well-presented and well supported by the literature.
6. Methods: The PRISMA guidelines are well-explained.
7. Results: Finding 17 studies represents a strong body of work. However, why include the three studies that were descriptive only? Perhaps mention these in the introduction or elsewhere; but it dilutes the quality of the systematic review. Response: We have renamed the article to clearly define the review as an integrative review, combing both empirical and other conceptual or descriptive articles.
8. Discussion: Generally good. Response: This has been updated as recommended by the other reviewer.
9. Conclusions and Limitations: Appropriate.
10. APA7 References: Good but a few random errors. Add digital object identifiers (dois) for journal articles. This is inconsistent. Response: The reference list has been edited and updated.